# YOLO-LWNet: A Lightweight Road Damage Object Detection Network for Mobile Terminal Devices

**DOI:** 10.3390/s23063268

**Published:** 2023-03-20

**Authors:** Chenguang Wu, Min Ye, Jiale Zhang, Yuchuan Ma

**Affiliations:** National Engineering Research Center of Highway Maintenance Equipment, Chang’an University, Xi’an 710065, China

**Keywords:** road damage detection, object detection, lightweight network, mobile terminal, YOLOv5, attention mechanism

## Abstract

To solve the demand for road damage object detection under the resource-constrained conditions of mobile terminal devices, in this paper, we propose the YOLO-LWNet, an efficient lightweight road damage detection algorithm for mobile terminal devices. First, a novel lightweight module, the LWC, is designed and the attention mechanism and activation function are optimized. Then, a lightweight backbone network and an efficient feature fusion network are further proposed with the LWC as the basic building units. Finally, the backbone and feature fusion network in the YOLOv5 is replaced. In this paper, two versions of the YOLO-LWNet, small and tiny, are introduced. The YOLO-LWNet was compared with the YOLOv6 and the YOLOv5 on the RDD-2020 public dataset in various performance aspects. The experimental results show that the YOLO-LWNet outperforms state-of-the-art real-time detectors in terms of balancing detection accuracy, model scale, and computational complexity in the road damage object detection task. It can better achieve the lightweight and accuracy requirements for object detection for mobile terminal devices.

## 1. Introduction

Damage to the road surface poses a potential threat to driving safety on the road; maintaining excellent pavement quality is an important prerequisite for safe road travel and sustainable traffic. The inability to detect road damage promptly is often considered one of the most critical steps in limiting the need to maintain high pavement quality. To maintain high pavement quality, transportation agencies are required to regularly evaluate pavement conditions and maintain them promptly. These road damage assessment techniques have gone through three stages in the past.

Manual testing was used in the earliest days, where the evaluator visually detects road damage by walking on or along the roadway. In the process of testing, evaluators need expertise in related fields and make field trips, which are tedious, expensive, and unsafe [1]. Moreover, this detection method is inefficient, and the road needs to be closed during on-site detection, which may affect the traffic flow. Therefore, manual testing technology is not appropriate for application to high-volume road damage detection. Gradually, semiautomatic detection technology became the primary method. In this method, images are automatically collected by vehicles traveling at high speeds, which requires specific equipment to capture the road conditions and reproduce the road damage. Then, the professional technicians identify the type of road damage and output the results manually [2,3,4]. This method greatly reduces the impact on traffic flow, but has some disadvantages, such as a heavy follow-up workload, a single detection target, and a high equipment overhead. Over the past 20 years, with the development of sensor technology and camera technology, the technology of integrated automatic detection of road damage has made great progress. Established automatic detection systems include vibration-based methods [5], laser-scanning-based methods [6,7], and image-based methods. Among them, vibration-based methods and laser-scanning-based methods require expensive specialized equipment. Moreover, the reliability of the system for road damage detection is highly dependent on the operator [8]. Therefore, although the above methods possess high detection accuracy, they are not practical when considering economic issues. The most highlighted features of image-based methods are their low cost and not needing special equipment [9,10,11,12,13,14,15,16]. However, image-based methods do not have the accuracy of vibration-based methods or laser-scanning-based methods, so image-based automated detection of road damage requires highly intelligent and robust algorithms.

The advent of machine learning has opened new directions in road damage detection with pioneering applications in pavement crack classification [17,18,19]. However, these studies only looked at the characteristics of shallow networks, which could not detect the complex information about the pavement, and could not distinguish between road damage categories. The unprecedented development of computer computing power has laid the foundation for the emergence of deep learning, which can effectively address the low accuracy of image-based methods of road damage detection. Deep learning has gained widespread attention in smart cities, self-driving vehicles, transportation, medicine, agriculture, finance, and other fields [20,21,22,23,24,25]. Deep learning has the following advantages over traditional machine learning: deep learning can train models directly using data without pre-processing; and deep learning has a more complex network composition and outperforms traditional machine learning in terms of feature extraction and optimization. From a certain point of view, we can consider road damage detection as an object detection task, which focuses on classifying and localizing the objects to be detected. In the object detection task, the deep learning model has a powerful feature extraction capability, which is significantly better than the manually set feature detectors. Deep learning-based object detection algorithms are divided into two main categories, including two-stage detection algorithms and one-stage detection algorithms. The former algorithm first finds the suggested regions from the input image, and then classifies and regresses each suggested region. Typical examples of this class of algorithms are R-CNN [26], Fast R-CNN [27], Faster R-CNN [28], and SPP-Net [29]. However, for the latter algorithm, there is no need to predict regions in advance, and the class probability and location coordinate values can be generated directly. Typical examples of this class of algorithms are the YOLO series [30,31,32,33,34,35], SSD [36], and RetinaNet [37].

In recent years, many studies have applied neural network models to pavement measurement or damage detection. References [38,39,40] used a neural network model to detect cracks in pavement. However, these road damage detection methods are only concerned with determining whether cracks are present. Based on this flaw, ref. [41] used a mobile device to photograph the road surface, divided the Japanese road damage into eight categories, named the collected dataset RDD-2018, and finally applied it to a smartphone for damage detection. After that, some studies focused on adding more images or introducing entirely new datasets [42,43,44,45,46], but the vast majority of these datasets were limited to road conditions in one country. For this purpose, in 2020, ref. [47] combined the road damage dataset from the Czech Republic and India with the Japanese road damage dataset [48] to propose a new dataset, “Road Damage Dataset-2020 (RDD-2020)”, which contains 26,620 images, nearly three times more than the 2018 dataset. In the same year, the Global Road Damage Testing Challenge (GRDDC) was held, in which several representative detection schemes were used [49,50,51,52,53,54]. The above studies found that object detection algorithms from other domains can also be applied to road damage detection tasks. However, as the performance evaluation of the competition is only F1-Score, the model inevitably increases its scale while improving its detection accuracy, so that the model occupies a large savings space, increases the inference time of the model, and has high requirements for equipment.

In road damage detection, mobile terminal devices are more suitable for detection tasks due to the limitations of the working environment. However, mobile terminal devices have limited storage capacity and computing power, and the imbalance between accuracy and complexity makes it difficult to apply the models on mobile terminal devices. In addition, state-of-the-art algorithms were trained from datasets such as Pascal VOC and COCO, not necessarily for road damage detection. Considering these shortcomings, in this study, the YOLO-LWNet algorithm, which balances detection accuracy and algorithm complexity, is proposed and applied to the RDD-2020 road damage dataset. In order to maintain high detection accuracy while effectively reducing model scale and computational complexity, we designed a novel lightweight network building block, the LWC, which includes a basic unit and a unit for spatial downsampling. According to the LWC, a lightweight backbone network suitable for road damage detection was designed for feature extraction, and the backbone network in the YOLOv5 was replaced by the lightweight network designed by us. A more effective feature fusion network structure was designed to achieve efficient inference while maintaining a better feature fusion capability. In this paper, we divide the YOLO-LWNet model into two versions, tiny and small. Comparing the YOLO-LWNet with state-of-the-art detection algorithms, our algorithm can effectively reduce the scale and computational complexity of the model and, at the same time, achieve a better detection result. 

The contribution of this study is as follows: ① To balance the detection accuracy, model scale, and computational complexity, a novel lightweight network building block, the LWC, was designed in this paper. This lightweight module can effectively improve the efficiency of the network model, and also effectively avoid gradient fading and enhance feature reuse, thus maintaining the accuracy of object detection. The attention module was applied in the LWC module, which re-weights and fuses features from the channel dimension. The weights of valid features are increased and the weights of useless features are suppressed, thus improving the ability of the network to extract features; ② A novel lightweight backbone network and an efficient feature fusion network were designed. When designing the backbone, to better detect small and weak objects, we expanded the LWC module to deepen the thickness of the shallow network in the lightweight backbone, to maximize the attention to the shallow information. To enhance the ability of the network to extract features of different sizes, we also used the Spatial Pyramid Set-Fast (SPPF) [36] module. In the feature fusion network, we adopted the topology based on BiFPN [55], and replaced the C3 module in the YOLOv5 with a more efficient structure, which effectively reduces the model scale and computational complexity of the network, while ensuring almost the same feature fusion capability; ③ To evaluate the effect of each design on the network model detection performance, we conducted ablation experiments on the RDD-2020 road damage dataset. The network model YOLO-LWNet, designed in this paper, is also compared with state-of-the-art object detection models on the RDD-2020 dataset. Through comparison, it is found that the network model designed in this paper improves the detection accuracy to a certain extent, effectively reduces the scale and computation complexity of the model, and can better achieve the deployment requirements of mobile terminal devices.

The rest of this paper is organized as follows. In Section 2, we will introduce the development of lightweight networks and the framework of the YOLOv5. In Section 3, the structural details of the lightweight network YOLO-LWNet will be introduced. The algorithm can effectively balance detection accuracy and model complexity. In Section 4, we will present the specific experimental results of this paper and compare them with the detection results of state-of-the-art methods. Finally, in Section 5, we will conclude this paper and propose some future works.

## 2. Related Work on The YOLO Series Detection Network and Lightweight Networks

### 2.1. Lightweight Networking

The deployment of the object detection network in mobile terminal devices mainly requires considering the scale and computational complexity of the model. The model minimizes the model scale and computational complexity while maintaining high detection accuracy. To achieve this goal, the common practices are network pruning [56], knowledge distillation [57], and lightweight networking. The network pruning technique is to optimize the already designed network model by removing the redundant weight channels in the model, and the compressed neural network model can achieve faster operation speed and lower computational cost. In knowledge distillation, ground truth is learned through a complex teacher model, followed by a simple student model that learns the ground truth while learning the output of the teacher model, and finally, the student model is deployed on the terminal device.

However, to be more applicable to mobile scenarios, more and more studies have directly designed lightweight networks for mobile scenarios, such as MobileNet [58,59,60], ShuffleNet [61,62], GhostNet [63], EfficientDet [55], SqueezeNet [64], etc. These networks have introduced some new network design ideas that can maintain the accuracy of model detection and effectively reduce the model scale and computational complexity to a certain extent, which is important for the deployment of mobile terminals. In addition to these manually designed lightweight networks, lightweight network models can also be automatically generated by computers, which is called Neural Architecture Search (NAS) [65] technology. In some missions, NAS technology could design on par with human experts, and it discovered many network structures that humans have not proposed. However, NAS technology consumes huge computational resources and requires powerful GPUs and other hardware as the foundation.

Among the manually designed lightweight networks, the MobileNet series and ShuffleNet series as the most representative lightweight network structures, have a significant effect on the reduction of the computational complexity and scale of the network model. The core idea of this efficient model is the depthwise separable convolution, which mainly consists of a depthwise (DW) convolution module and a pointwise (PW) convolution module. Compared with the conventional convolution operation, this method has a smaller model size and lower computational costs. The MobileNetv3 employs a bottleneck structure while using the depthwise separable convolution principle. The structure is such that the input is first expanded in the channel direction, and then reduced to the size of the original channel. In addition, the MobileNetv3 also introduces the specification-and-excitation (SE) module, which improves the quality of neural network feature extraction by automatically learning to obtain the corresponding weight values of each feature channel. Figure 1 shows the structure of the MobileNetv3 unit blocks. First, the input is expanded by 1 × 1 PW in the channel direction, and then the feature map after the expansion is extracted using 3 × 3 DW, then the corresponding weight values of the channels are learned by SE, and, finally, the feature maps are reduced in the channel direction. If we set the expanded channel value to the input channel value, no channel expansion operation will be performed, as shown in Figure 1a. To reduce computational costs, MobileNetv3 uses hard-sigmoid and hard-swish activation functions. When the stride in 3 × 3 DW is equal to one and the input channel value is equal to the output channel value, there will be a residual connection in the MobileNetv3 unit blocks, as shown in Figure 1b.

The ShuffleNetv2 considers not only the impact of indirect indicators of computational complexity on inference speed, but also memory access cost (MAC) and platform characteristics. The structures of the ShuffleNetv2 unit blocks are shown in Figure 2. Among them, Figure 2a shows the basic ShuffleNetv2 unit, and the feature map is operated by a channel split at the input, so that the input feature map is equally divided into two parts in the direction of the channel. One of the parts does not change anything, and the other part performs a depthwise separable convolution operation. Unlike the MobileNetv3, there is no bottleneck structure in the ShuffleNetv2, where the 1 × 1 PW and 3 × 3 DW do not change the channel value for the input. Finally, a concat operation is performed on both parts, and a channel shuffle operation is performed to enable information interaction between the different parts. Figure 2b shows the network structure of the spatial downsampling unit of the ShuffleNetv2, removing the channel shuffle module and performing 3 × 3 DW and 1 × 1 PW operations on the unprocessed branch in the basic unit. Eventually, the resolution of the output feature map is reduced to half the resolution of the input feature map, and the feature map channel value is increased to some extent.

Although the existing lightweight networks can effectively reduce the scale and computational complexity of detection models, there is still a need for further research in terms of performance, such as detection accuracy. In the application of road damage detection tasks, the detection model using the ShuffleNetv2 and the MobileNetv3 as the backbone can effectively reduce the scale and computational complexity of the models, but at the significant expense of detection effectiveness. This paper combined the design ideas of the existing lightweight networks and designed a novel LWC lightweight module, and then compared the performance with the ShuffleNetv2 and the MobileNetv3 on the RDD-2020 dataset.

### 2.2. YOLOv5

The YOLO family applies CNN for image detection, segments the image into blocks of equal size, and make inferences on the category probabilities and bounding boxes of each block. The inference speed of the YOLO is so fast that it is nearly 1000 times faster than R-CNN inference, and 100 times faster than Fast R-CNN inference. Based on the original YOLO model, the YOLOv5 references many optimization strategies in the field of CNN in the network. Currently, the YOLOv5 has officially been updated to the v6.1 version. The YOLOv5 backbone mainly implements the extraction of feature information in the input image through the focus structure and CSPNet [66]. However, in the v6.1 version, a 6 × 6 convolution is used instead of the focus structure to achieve the same effect. The YOLOv5 uses the PANet [67] structural fusion layer to fuse the three features extracted from the backbone at different resolutions. The YOLOv5 object detection network model consists of four architectures, named YOLOv5-s, YOLOv5-m, YOLOv5-l, and YOLOv5-x. The main difference between them is the different depth and width of each model. In this paper, we comprehensively considered the detection accuracy, inference speed, model scale, and computational complexity of the object detection model, and designed the novel road damage detection model based on the YOLOv5-s. Figure 3 shows the structure of the YOLOv5-s.

The network structure of the YOLOv5-s consists of four modules: The first part is the input, which is the image, and the resolution size of the input is 640 × 640. The resolution size of the image is scaled as required by the network settings and performs operations, such as normalization. When training the model, the YOLOv5 uses mosaic data enhancement to improve the accuracy of the network, and proposes an adaptive anchor frame calculation. The second part is the backbone, which is usually a well-performing classification network, and this part is used to extract common feature information. The YOLOv5 uses the CSPDarknet53 to extract the main feature information in the image. The CSPDarknet53 is composed of focus, CBL, C3, SPP, and other modules, and their structure is shown in Figure 3. The focus module splits the images to be detected along landscape and portrait orientation, and then concats it along the channel direction. In v6.1, 6 × 6 convolution replaces the focus module. In the backbone, C3 consists of n Bottleneck (residual structures), SPP performs max-pooling of three different sizes on the inputs and concats them. The third part is the neck, usually located between the backbone and the head, which can effectively improve the robustness of the extracted features. The YOLOv5 fuses the information of different network depth features through the PANet structure. The fourth part is the head, which is used to output the results of object detection. The YOLOv5 implements predictions on three different sizes of feature maps.

The improvement of detection accuracy in the YOLOv5 makes the network more practical in object detection, but the memory size and computing power of mobile terminal devices are limited, making it difficult to deploy road damage object detection with good detection results. In this paper, we designed a novel lightweight road damage detection algorithm, the YOLO-LWNet, which is mainly applicable to mobile terminal devices. The experimental results validated the effectiveness of the novel road damage detection model. A balance between detection accuracy, model scale, and computational complexity is effectively achieved, which provides the feasibility of deploying the network model on the mobile terminal device.

## 3. Proposed Method for Lightweight Road Damage Detection Network (YOLO-LWNet)

### 3.1. The Structure of YOLO-LWNet

The YOLO-LWNet is a road damage detection network applied to mobile terminal devices. Its structure is based on the YOLOv5. Recently, many YOLO detection frameworks have been derived, among which the YOLOv5 and YOLOX [68] are the most representative. However, in practical applications for road damage detection, they have considerable computational costs and large model scale, which make them unsuitable for application on mobile terminal devices. Therefore, this paper develops and designs a novel road damage detection network model, named YOLO-LWNet, through the steps of network structure analysis, data division, parameter optimization, model training, and testing. The network model mainly improves the YOLOv5-s in terms of backbone, neck, and head. The structure of the YOLO-LWNet-Small is shown in Figure 4, which mainly consists of a lightweight backbone for feature extraction, a neck for efficient feature fusion, and a multi-scale detection head.

In Figure 4, after the focus operation, the size of the input tensor is changed from 640 × 640 × 3 to 320 × 320 × 32. Then, the information extraction of the feature maps with different depths is accomplished using the lightweight LWC module. An attention mechanism module is inserted in the LWC module to guide different weight assignments to extract weak features. In this paper, the ECA [69] and CBMA [70] attention modules are introduced in the LWC module to enhance feature extraction. Subsequently, the SPPF module is used to avoid the problem of image information loss during scaling and cropping operations on the image area, greatly improving the speed of generating candidate frames, and saving computational costs. Then, in the neck network architecture, this paper adopts a BiFPN weighted bidirectional pyramid structure instead of PANet to generate feature pyramids, and adopts bottom-up and top-down fusion to effectively fuse the features extracted by the backbone at different resolution scales, and improve the detection effect at different scales. Moreover, three sets of output feature maps with different resolution sizes are detected in the detection head. Ultimately, the neural network generates a category probability score, a confidence score, and the coordinate values of the bounding box. Then, the detection results are screened according to the post-processing of NMS to obtain the final object detection results. In this paper, we also have another version of the object detection model, the YOLO-LWNet-Tiny, whose network model can be obtained by adjusting the width factor in the small version. Compared to state-of-the-art object detection algorithms, in terms of road damage detection, the YOLO-LWNet model has qualitative improvements in detection accuracy, model scale, and computational complexity.

### 3.2. Lightweight Network Building Block—LWC

To reduce the model scale and computational complexity of network models, a novel lightweight network construction unit is designed for small networks in this paper. The core idea is to reduce the scale and computational cost of the model using depthwise separable convolution, which consists of two parts: depthwise convolution and pointwise convolution. Depthwise convolution performs individual convolution filtering for each channel of the feature map, which can effectively lighten the network. Pointwise convolution is responsible for information exchange between feature maps in the channel direction, and a linear combination of channels is performed to generate new feature maps. Assuming that a feature map of size *C_P_* × *C_P_* × *X* is given to the standard convolution input, and an output feature map of size *C_F_* × *C_F_* × *Y* is obtained, and the size of the convolution kernel is *C_K_* × *C_K_* × *X*, then the computational cost of the standard convolution to process this input is *C_K_* × *C_K_* × *X* × *Y* × *C_F_* × *C_F_*. Using the same feature map as the input of depthwise separable convolution, suppose there are *X* depthwise convolution kernels of size *C_K_* × *C_K_* × 1, and *Y* pointwise convolution kernels of size 1 × 1 × *X*, the computational cost of depthwise separable convolution is *C_K_* × *C_K_* × *X* × *C_F_* × *C_F_* + *X* × *Y* × *C_F_* × *C_F_*. After comparing their computational costs, it can be seen that the computational cost of depthwise separable convolution is $\frac{1}{Y}+\frac{1}{C_{K}^{2}}$ times than that of the standard convolution. Since $\frac{1}{Y}+\frac{1}{C_{K}^{2}}<1$, the network model can be kept computationally low by a reasonable application of depthwise separable convolution. Suppose a 640 × 640 × 3 image is fed to the convolutional layer and a 320 × 320 × 32 feature map is obtained. When the convolutional layer uses 32 standard convolutional kernels of size 3 × 3 × 3, the computation is 884,736,000. When using depthwise separable convolution, the computation is 125,952,000. The computation is reduced by a factor of seven, which shows that depthwise separable convolution can reduce the computation of the model very well, and the deeper the network, the more effective it is.

The lightweight network building block designed in this paper ensures that more features are transferred from the shallow network to the deep network, which promotes the transfer of gradients. Specifically, the LWC module includes two units; one is a basic unit and the other is a unit for spatial downsampling. Figure 5a shows the specific network structure of the basic unit. To better communicate the feature information between each channel, we use channel splitting and channel shuffling at the beginning and end of the basic unit, respectively. In the beginning, the feature map is split into two branches of an equal number of channel dimensions using the channel split operation. One branch is left untouched, and the other branch uses depthwise separable convolutions for processing. Specifically, the feature map is first expanded in the channel direction using a 1 × 1 pointwise convolution with an expansion rate of R. This operation can effectively avoid the loss of information. Then, the 2D feature maps on the n channels are processed one by one using n 3 × 3 depthwise convolutions. Finally, the dimension is reduced to the size before decoupling through 1 × 1 pointwise convolution, after which the attention mechanism module is inserted to achieve the extraction of weak features. In the whole architecture, activation tensors of corresponding dimensions are added after each 1 × 1 pointwise convolution operation. After the operations of the above two branches, the channel concatenation operation is used to merge the information of the two branches, and the channel shuffle is used to avoid blocking the information ground exchange between the two branches. Three consecutive element-wise operations channel concatenation, channel split, and channel shuffle are combined into one element-wise operation. For the unit for spatial downsampling, to avoid the reduction of parallelism caused by network fragmentation, the unit for spatial downsampling only adopts the structure of a single branch. Although a fragmented structure is good for improving accuracy, it reduces efficiency and introduces additional overhead. As shown in Figure 5b, compared to the basic unit, the module unit removes the channel split operator when spatial downsampling. The feature map is directly subjected to a decompressed 1 × 1 PW, a 3 × 3 DW, with stride set to two, a compressed 1 × 1 PW, an attention module, and a channel shuffling operator. Finally, the unit for spatial downsampling is processed so that the resolution size of the feature map is reduced to half of the original feature map, and the number of channels is increased to *N* (*N* > 1) times the original number of channels.

### 3.3. Attention Mechanism

The YOLOv5 is applied to road damage detection, where the road damages in the image are treated equally. If a certain weight coefficient is assigned to the feature map of the object region during object detection, this weighting method is beneficial to improve the attention of the network model for object detection, and this method is known as an attention mechanism. SENet [71] is a common channel attention mechanism. As shown in Figure 6a, its structure consists of a global average pooling layer, two fully connected layers, and a sigmoid operation. SENet outputs the weight value corresponding to each channel, and finally, the weight value is multiplied by the feature map to complete the weighting operation. Using SENet will bring certain side effects, which increase the inference time and model scale. To reduce the aforementioned negative effects, in Figure 6b, ECANet improves SENet by removing the fully connected layer from the network and learning the features through a 1D convolution, where the size of the 1D convolution kernel affects the coverage of cross-channel interactions. In this paper, we chose the 1D convolution kernel size as three. The ECANet only applies a few basis parameters on SENet and achieves obvious performance gains.

In the LWC block, ECA is used in the basic unit for channel attention, while CBAM [70] is used in the unit for spatial downsampling to combine channel attention and spatial attention. The network structure of CBAM is shown in Figure 7, which processes the input feature maps with the channel attention mechanism and the spatial attention mechanism, respectively. The channel attention of CBAM uses both the global maximum pooling layer and global average pooling layer to effectively calculate the weight values of channels, and generates two different spatial descriptor vectors: $F_{max}^{c}$ and $F_{avg}^{c}$. Then, the two vectors are generated into a channel attention vector *M_c_* through a network of shared parameters. The shared network is composed of a multi-layer perceptron (*MLP*) and a hidden layer. To reduce the computational cost, the pooled channel attention vectors are reduced by *r* times, and then activated. In short, the channel attention can be represented by Formula (1):(1)$$\;M_{c}\left(F\right)=\sigma (MLP\left(AvgPool\left(F\right)\right)+MLP\left(MaxPool\left(F\right)\right)\;=\sigma \left(W_{1}\left(W_{0}\left(F_{avg}^{c}\right)\right)+W_{1}\left(W_{0}\left(F_{max}^{c}\right)\right)\right)$$
where *σ* is the sigmoid function, and the weights *W*_0_ and *W*_1_ of the *MLP* are shared in the two branches.

The spatial attention module takes advantage of the relationships between feature maps to the spatial attention feature map. First, the average pooling and maximum pooling operations are performed on the feature map along the channel direction, and the two results are concatenated to generate a feature map with a channel number of two. Then, a convolutional layer is applied to generate a 2D spatial attention map. Formula (2) represents the calculation process of the spatial attention:(2)$$\;M_{s}\left(F\right)=\sigma \left(f^{7\times 7}\left(\left[AvgPool\left(F\right);MaxPool\left(F\right)\right]\right)\right)\;=\sigma \left(f^{7\times 7}\left(\left[F_{avg}^{s};F_{max}^{s}\right]\right)\right)\;\;\;\;\;\;\;\;\;\;\;\;\;\;\;\;\;$$
where σ denotes the sigmoid function and $f^{7\times 7}$ represents the convolution layer with the size of 7 × 7 for this convolutional kernel. $F_{avg}^{s}$ and $F_{max}^{s}$ denote the feature maps generated by average pooling and maximum pooling. In CBAM, there are two main hyperparameters, which are the dimensionality reduction multiple *r* in the channel attention, and the convolutional layer convolution kernel size *k* in the spatial attention. In this study, the officially specified hyperparameter values are chosen: *r* = 16, and *k* = 7.

### 3.4. Activation Function

In the backbone network of the YOLO-LWNet, we introduced a nonlinear function, hardswish [60], an improved version of the recent swish [72] nonlinear function, which is faster in calculation and more friendly to mobile terminal devices. While the activation function swish in Formula (3) allows the neural network model to have high detection, it is costly to calculate the sigmoid function on mobile terminal devices. This limits the adoption of deep learning network models in mobile terminal devices. Based on this problem, we use the hardswish function instead of the swish function, and the sigmoid function is replaced by ReLU6(*x* + 3)/6, as seen in Formula (4). Through experiments, we found that the hardswish function makes no significant difference in the detection effect, but can improve the detection speed.
(3)$$\mathrm{swish}\left(x\right)=x\times \sigma \left(x\right)$$
(4)$$\mathrm{hardswish}\left(x\right)=x\times \frac{ReLU6\left(x+3\right)}{6}$$

### 3.5. Lightweight Backbone—LWNet

By determining the attention mechanism and the activation function, the LWC module required to construct the backbone can be obtained. These building blocks are repeatedly stacked to build the entire backbone. In Figure 8, after the focus operation of the input image, four spatial downsampling operations are performed in the backbone, respectively, and these spatial downsamplings are all implemented by the LWC unit for spatial downsampling. LWB is inserted after each spatial downsampling unit, as shown in Figure 8, and each LWB is composed of *n* basic LWC units. The size of *n* determines the depth of the backbone, and the width of the backbone is determined by the size of the output feature map of each layer in the channel direction. Finally, the SPPF layer is introduced at the end of the backbone. The SPPF layer can pool feature maps of any size and generate a fixed-size output, which can avoid clipping or deformation of the shallow network; its specific structure is the same as SPPF in the YOLOv5.

When designing the backbone, we need to determine the hyperparameters, including the expansion rate *R* and the number of channels in the building block, as well as the number of units of LWC contained in each layer of the LWblock. With the gradual deepening of network layers, the deep layers can better extract high-level semantic information, but it has a lower resolution of the feature map. In contrast, the feature map resolution of shallow layers is high, but the shallow network is weak in extracting high-level semantic information. In road damage detection tasks, most of the detected object categories are cracks, which may be only a few pixels wide, and the information from the deep layers of the network may lose the visual details of these road damages. To adequately extract the features of road damage, it is required that the neural network can extract excellent high-resolution features at shallow layers. Therefore, in the backbone, the width and depth of the shallow LWC module are increased to realize multi-feature extraction from each network depth. In the backbone, we use a nonconstant expansion rate in all layers, except the first and last layers. Its specific values are given in Table 1. For example, in the LWNet-Small model, for the LWConv (stride set to two) layer, when it inputs a 64-dimensional channel count feature map and produces a 96-dimensional channel count feature map, then the intermediate expansion layer is 180-dimensional channel count; for the LWConv (stride set to one) layer, it inputs a 96-channel feature and generates a 96-channel feature, with 180 channels for the intermediate extension layer. In addition, the width, depth factors, and expansion rate of the backbone are controlled as tunable hyperparameters to customize the required architecture for different performance, and adjust it according to the desired accuracy/performance trade-offs. Specifically, the lightweight backbone, LWNet, is defined as two versions: LWNet-Small and LWNet-Tiny. Each of these versions is intended for use with different resource devices. The full parameters of the LWNet are shown in Table 1. We determined the specific width, depth factors, and expansion rate of the LWNet through experiments.

### 3.6. Efficient Feature Fusion Network

In terms of neck design, we want to achieve a more efficient reasoning process and achieve a better balance between detection accuracy, model scale, and computational complexity. This paper designs a more effective feature fusion network structure based on the LWC module. Based on BiFPN topology, this feature fusion network replaces the C3 module used in the YOLOv5 with the LWblock, and adjusts the width and depth factors in the overall neck. This achieves an efficient inference on hardware, and maintains good multi-scale feature fusion capability; its specific structure is shown in Figure 9. The features extracted from the backbone are classified into several types according to the resolution, and are noted as P1, P2, P3, P4, P5, etc. The numbers in these markers represent the number of times the resolution was halved. Specifically, the resolution of P4 is 1/16 of the input image resolution. The feature fusion network in this paper fuses P3, P4, and P5 feature maps. The neck LWblock differs from the backbone LWblock in its structure; the neck LWblock is composed of a 1 × 1 Conv and *N* basic units of the LWC module. When the input channels in the LWblock are different from the output channels, the number of channels is first changed using the 1 × 1 Conv, and then the processing continues using the LWConv. The specific parameters of each LWblock are given in Figure 9.

## 4. Experiments on Road Damage Object Detection Network

In this paper, the novel road damage object detection network YOLO-LWNet was trained using the RDD-2020 dataset, and its effectiveness was verified. Firstly, we compared the advanced lightweight object detection algorithms with the RDD-mobilenet designed in this paper, in terms of detection accuracy, inference speed, model scale, and computational complexity. Secondly, ablation experiments were conducted to test the ablation performance resulting from different improved methods. Finally, to prove the performance improvement of the final lightweight road object detection models, they were compared with state-of-the-art object detection algorithms.

### 4.1. Datasets

The dataset used in this study was proposed in [73], and includes one training set and two testing sets. The training set consists of 21,041 labeled road damage images, which include four damage types obtained through intelligent device collection from Japan, the Czech Republic, and India. The damage categories include longitudinal crack (D00), transverse crack (D10), alligator crack (D20), and potholes (D40). Table 2 shows the four types of road damage and the distribution of road damage categories in the three countries, with Japan having the most images and damage, India having fewer images and damage, and the Czech Republic having the least number of images and damage. In the RDD-2020 dataset, its test set labels are not available, and to run the experiments, we divided the training dataset into a training set, a validation set, and a test set in the ratio of 7:1:2.

### 4.2. Experimental Environment

In our experiments, we used the following hardware environment: the GPU is NVIDIA GeForce RTX 3070ti, the CPU is Intel i7 10700k processor, and the memory size is 32GB. The software environment: Windows 10 operating system, Python 3.9, CUDA 11.3, cuDNN 8.2.1.32, and PyTorch 1.11.0. In the experiments, the individual network models were trained from scratch by using the RDD-2020 dataset. To ensure fairness, we trained the network models in the same experimental environment, and validated the detection performance of the trained models on the test set. To ensure the reliability of the training process, the hyperparameters of the model training were kept consistent throughout the training process. The specific training hyperparameters were set as follows: the input image size in the experiment was 640 × 640, the epochs of the entire training process were set to 300, the warmup epochs were set to 3, the batch size was 16 during training, the weight decay of the optimizer was 0.0005, the initial learning rate was 0.01, the loop learning rate was 0.01, and the learning rate momentum was 0.937. In each epoch, mosaic data enhancement and random data enhancement were turned on before training.

### 4.3. Evaluation Indicators

In deep learning object detection tasks, detection models are usually evaluated in terms of recall, precision, average precision rate (*AP*), mean average precision rate (*mAP*), params, floating-point operations (FLOPs), frames per second (FPS), latency, etc.

In object detection tasks, the precision indicator cannot directly evaluate for object detection. For this reason, we introduced the *AP*, which calculates the area under a certain type of P-R curve. When calculating *AP*, it is first necessary to calculate precision and recall, precision measures the percentage of correct predictions among all results predicted as positive samples, and recall measures the percentage of correct predictions among all positive samples. For the object detection task, an *AP* value can be calculated for each category, and the *mAP* is the average of the *AP* values of all categories. The expressions corresponding to precision, recall, and *mAP* are shown in Formula (5), Formula (6), and Formula (7), respectively:(5)$$P=\frac{T_{P}}{T_{P}+F_{P}}$$
(6)$$R=\frac{T_{P}}{T_{P}+F_{N}}$$
(7)$$mAP=\frac{\sum_{i=1}^{n}AP_{i}}{n}$$

To compute precision and recall, as with all machine learning problems, true positives (*T_P_*), false positives (*F_P_*), true negatives (*T_N_*), and false negatives (*F_N_*) need to be determined. Where *T_P_* indicates that the test result for a positive sample is a positive sample; *F_P_* indicates that the test result for a negative sample is a positive sample; and *F_N_* indicates that the test result for a positive sample is a negative sample. When calculating precision and recall, IoU and confidence thresholds have a great influence on them, and can directly affect the change of the P-R curve. In this paper, we took the IoU value as 0.6 and the confidence threshold as 0.1.

In addition to accuracy, computational complexity is another important consideration. Engineering applications are often designed to achieve the best accuracy with limited computing resources, which is caused by the limitations of the target platform and application scenarios. To measure the complexity of a model, a widely used indicator is FLOPs, and another indicator is the number of parameters, denoted by params. Where FLOPs are used to represent the theoretical computation of the model, the unit of the large model is usually G, and the unit of the small model is usually M. Params relate to the size of the model file, usually in M. 

However, FLOPs is an indirect indicator. The detection speed is another important evaluation indicator for object detection models, and only a fast speed can achieve real-time detection. In previous studies [74,75,76,77], it has been found that networks with the same FLOPs have different inference speeds. Because FLOPs only consider the amount of computation of the convolutional part, although this part takes up most of the time, other operations (such as channel shuffling and element operations) also take up considerable time. Therefore, we used FPS and latency as direct indicators, while using FLOPs as indirect indicators. FPS is the number of images processed in one second, and the time required to process an image is the latency. In this paper, we tested different network models on a GPU (NVIDIA GeForce RTX 3070ti). FP16-precision and batch set to one were used for measurement during the tests.

### 4.4. Comparison with Other Lightweight Networks

To verify the performance of the lightweight network unit LWC in road damage detection, we used the MobileNetV3-Small, the MobileNetV3-Large, the ShuffleNetV2-x1, and the ShuffleNetV2-x2 as substitutes for backbone feature extraction network in the YOLOv5. We compared them with the lightweight network based on the LWC module (BLWNet) unit on the RDD-2020 public dataset. Specifically, we used the MobileNetV3-Small, the ShuffleNetV2-x1, and the BLWNet-Small as the backbone of feature extraction for the YOLOv5-s model. The MobileNetV3-Large, the ShuffleNetV2-x2, and the BLWNet-Large were used as the backbone of the YOLOv5-l model. To fairly compare the performance of each structure in road damage detection, we selected the attention mechanism and activation functions used in the MobileNetV3 and the ShuffleNetV2 for the LWC module. In addition, we designed different versions of the model by adjusting the output channel value, the exp channel value, and the number of modules *n* of the LWC module. In the experiment of this stage, the specific parameters of the two sizes of models designed in this paper are shown in Table 3 and Table 4, respectively.

In the neck network of the YOLOv5, three resolution feature maps are extracted from the backbone network, and the three extracted feature maps are named P3, P4, and P5, respectively. P3 corresponds to the output of the last layer with a step of 8, P4 corresponds to the output of the last layer with a step of 16, and P5 corresponds to the output of the last layer with a step of 32. For the BLWNet-Large, P3 is the 7th LWConv layer, P4 is the 13th LWConv layer, and P5 is the last LWConv layer. For the BLWNet-Small, P3 is the sixth LWConv layer, P4 is the ninth LWConv layer, and P5 is the last LWConv layer.

We compared *mAP*, param, FLOPs, FPS, and latency, and in Table 5 the test results of each network model on the RDD-2020 test set are shown. As can be seen, the BLWNet-YOLOv5 is not only the smallest model in terms of size, but also the most accurate and fastest model among the three models. The BLWNet-Small is 0.8 ms faster than the MobileNetV3-Small and has a 2.9% improvement in *mAP*. Compared to the ShuffleNetV2-x1, the the BLWNet-Small has 1.3 ms less latency and 3.1% higher mAP. The *mAP* of the BLWNet-Large with increased channels is 1.1% and 3.1% higher than those of the MobileNetV3-Large and the ShuffleNetV2-x2, respectively, with similar latency. The BLWNet outperforms the MobileNetV3 and the ShuffleNetV2 in terms of the reduced model scale, reduced computational cost, and improved *mAP*. In addition, the BLWNet model with smaller channels has better performance in reducing the latency. Although the BLWNet has an excellent performance in reducing model scale and computational cost, its *mAP* still has great room for improvement, so our next experiments mainly focus on improving the *mAP* of the network model when tested on the RDD-2020 dataset.

### 4.5. Ablation Experiments

To investigate the effect of different improvement techniques on the detection results, we conducted ablation experiments on the RDD-2020 dataset. Table 6 shows all the schemes in this study; in the LW scheme, only the backbone of the YOLOv5-s was replaced with the BLWNet-Small. In the LW-SE scheme, the CBAM attention module replaced the attention in the basic unit, and the ECA attention module replaced the attention in the unit for spatial downsampling. In the LW-SE-H scheme, the hardswish nonlinear function was used instead of the swish in the LWC module. In the LW-SE-H-depth scheme, the numbers of LWConv layers between P5 to P4, P4 to P3, and P3 to P2 in the BLWNet-Small were increased. In the LW-SE-H-depth-spp scheme, the SPPF module was added to the last layer of the backbone. In the LW-SE-H-depth-spp-bi scheme, based on the LW-SE-H-depth-spp, the BiFPN weighted bi-directional pyramid structure was used to replace PANet, to generate a feature pyramid. In the LW-SE-H-depth-spp-bi-ENeck scheme, the C3 blocks in the neck of the YOLOv5 were replaced by the LWblock proposed in this paper, to achieve efficient feature fusion. In the experiments, we found that CBAM attention modules would cause considerable latency in the model. Therefore, the scheme of the LW-SE-H-depth-spp-bi-fast, based on the LW-SE-H-depth-spp-bi, replaced the CBAM attention modules in the basic unit of the LWC with the ECA attention modules. In addition, the value of the channel output by the focus module in the network was reduced from 64 to 32. For the LW-SE-H-depth-spp-bi-ENeck-fast scheme, different from the LW-SE-H-depth-spp-bi-fast, the CBAM attention modules in the basic unit of the LWC in the backbone and neck were replaced by ECA attention modules.

The results of the ablation experiments are shown in Figure 10. There is still room for improvement in the param, FLOPs, and *mAP* of the YOLOv5-s model. The LW scheme reduces the parameter size and computational complexity of the YOLOv5-s by nearly half. However, the mAP decreased by 2.1%. To improve the *mAP*, the LW-SE scheme increases the *mAP* by 0.7% and reduces the model scale, but increases the latency. The LW-SE-H replaces the activation functions with hardswish, which has a positive impact on the inference speed, without reducing the detection accuracy, or changing the model scale and computational complexity. The LW-SE-H-depth enhances the extraction ability of 80 × 80, 40 × 40, and 20 × 20 resolution features, respectively, and increases the *mAP* by 2.0%. While deepening the network, the model scale and computational complexity inevitably increased, and the inference time of 4.4 ms is sacrificed. By introducing the improved SPPF module in the LW-SE-H-depth-spp, the features of different sensitive fields can be extracted, and the *mAP* can be further improved by 1.0% with essentially the same param, FLOPs, and latency. The application of the BiFPN structure makes the *mAP* of the network increase by 0.3%, and hardly increases the param, FLOPs, and latency of the model. For the application of an efficient feature fusion network, the param, and FLOPs of the network model can be greatly reduced, and the latency of 6.9 ms is sacrificed. To better balance detection accuracy, model scale, computational complexity, and inference speed, the LW-SE-H-depth-spp-bi-fast scheme replaced the attention module in the LW-SE-H-depth-spp-bi, and reduced the output channel value of the focus module. In this case, the *mAP* is decreased by 0.2%, the FLOPs of the model are effectively reduced, and the inference speed is greatly increased. Compared with that of the LW-SE-H-depth-spp-bi-ENeck scheme, the *mAP* of the LW-SE-H-depth-spp-bi-ENeck-fast is reduced by 0.1%, and the model scale and computational complexity are optimized to some extent, and the latency time is reduced by 8.2 ms. Finally, considering the balance between param, FLOPs, *mAP*, and latency of the network model, we chose the scheme LW-SE-H-depth-spp-bi-fast as the small version of the road damage object detection model YOLO-LWNet proposed in this paper. For the tiny version, we just need to reduce the width factor in the model. The YOLO-LWNet-Small has advantages over the original YOLOv5-s in terms of model performance and complexity. Specifically, our model increases the *mAP* by 1.7% in the test set, and it has a smaller number of parameters, almost half that of the YOLOv5-s. At the same time, its computational complexity is much smaller than that of the YOLOv5-s, which makes the YOLO-LWNet network model more suitable for mobile terminal devices. Compared with the YOLOv5-s, the inference time of the YOLO-LWNet-Small is 3.3 ms longer; this phenomenon is mainly caused by the depthwise separable convolution operation. The depthwise separable convolution is an operation with low FLOPs and a high data read and write volume, which consumes a large amount of memory access costs (MACs) in the process of data reading and writing. Limited by GPU bandwidth, the network model wastes a lot of time in reading and writing data, which makes inefficient use of the computing power of the GPU. For the mobile terminal device with limited computing power, the influence of MACs can be ignored, so this paper mainly considers the number of parameters, computational complexity, and detection accuracy of the network model. 

To better observe the effects of different improvement methods on the network model during the training process, Figure 11 shows the *mAP* curves of the nine experimental scenarios training during the training process. The horizontal axis in the figure indicates the number of training epochs, and the vertical axis indicates the value of the *mAP*. As the number of training epochs increases, the *mAP* also increases until after 225 epochs, where the *mAP* of all network schemes reaches the maximum value and the network model starts to converge. The LW-SE-H-depth-spp-bi-fast is the final proposed network model in this paper. From the figure, we can see that each improvement scheme can effectively improve the detection accuracy of the network, and we can intuitively observe that the final network model is better trained than the original one.

### 4.6. Comparison with State-of-the-Art Object Detection Networks

The YOLO-LWNet is a road damage detection model based on the YOLOv5, which includes two versions, namely, small and tiny, and the specific parameters of their backbone (LWNet) have been given in Table 1. Specifically, the YOLO-LWNet-Small is a model that uses the LWNet-Small as the backbone of the YOLOv5-s, and uses BiFPN as the feature fusion network. The YOLO-LWNet-Tiny uses BiFPN as the feature fusion network while using the LWNet-Tiny as the backbone of the YOLOv5-n. Figure 12 shows the results of the YOLO-LWNet model and state-of-the-art object detection algorithms for the experiments on the RDD-2020 dataset.

Compared with the YOLOv5 and the YOLOv6, the YOLO-LWNet has a great improvement in various indicators, especially in *mAP*, param, and FLOPs. Compared with the YOLOv6-s, the YOLO-LWNet-Small has a 78.9% and 74.6% reduction in param and FLOPs, respectively, the *mAP* is increased by 0.8%, and the latency is decreased by 3.9 ms. Compared with the YOLOv6-tiny, the YOLO-LWNet-Small model has 62.6% less param, 54.8% fewer FLOPs, 0.9% more *mAP*, and 1.8 ms of latency savings. The YOLO-LWNet-Small has no advantage over the YOLOv6-nano in terms of inference speed and computational complexity, but it effectively reduces param by 15.8%, and increases *mAP* by 1.6%. Compared to the YOLOv5-s, the small version has 1.7% more *mAP*, 48.4% less param, and 30% fewer FLOPs. The YOLO-LWNet-Tiny is the model with the smallest model scale and computational complexity, as shown in Figure 12. Compared with the YOLOv5-nano, param decreased by 35.8%, FLOPs decreased by 4.9%, and *mAP* increased by 1.8%. The YOLO-LWNet has less model scale, lower computational complexity, and higher detection accuracy in road damage detection tasks, which can balance the inference speed, detection accuracy, and model scale. In Figure 13, Figure 14, Figure 15 and Figure 16, the YOLO-LWNet is compared with five other detection methods for each of the four objects, and the predicted labels and predicted values for these samples are displayed. From the observation of the detection results, it is easy to find that the YOLO-LWNet can accurately detect and classify different road damage locations; the results are better than other detection network models. This shows that our network model can perform the task of road damage detection better. Figure 17 shows the detection results of the YOLO-LWNet.

## 5. Conclusions

In road damage object detection, a mobile terminal device is more suitable for detection tasks due to the limitations of the working environment. However, the storage capacity and computing power of mobile terminals are limited. To balance the accuracy, model scale, and computational complexity of the model, a novel lightweight LWC module was designed in this paper, and the attention mechanism and activation function in the module were optimized. Based on this, a lightweight backbone and an efficient feature fusion network were designed. Finally, under the principle of balancing detection accuracy, inference speed, model scale, and computational complexity, we experimentally determined the specific structure of the lightweight road damage detection network, and defined it as two versions (small and tiny) according to the network width. In the RDD-2020 dataset, the model scale of the YOLO-LWNet-Small is decreased by 78.9%, the computational complexity is decreased by 74.6%, the detection accuracy is increased by 0.8%, and inference time is decreased by 3.9 ms compared with the YOLOv6-s. Moreover, compared with the YOLOv5-s, the model scale of the YOLO-LWNet-Small is reduced by 48.4%, the computational complexity is reduced by 30%, and the detection accuracy is increased by 1.7%. For the YOLO-LWNet-Tiny, it has a 35.8% reduction in model scale, a 4.9% reduction in computational complexity, and a 1.8% increase in detection accuracy compared to the YOLOv5-nano. Through experiments, we found that the YOLO-LWNet is more suitable for the requirements of accuracy, model scale, and computational complexity of mobile terminal devices in road damage object detection tasks. 

In the future, we will further optimize the network model to improve its detection accuracy and detection speed, and increase the training data to make the network model more competent for the task of road damage detection. We will deploy the network model to mobile devices, such as smartphones, so that it can be fully used in the engineering field.

## Figures and Tables

**Figure 1 sensors-23-03268-f001:**
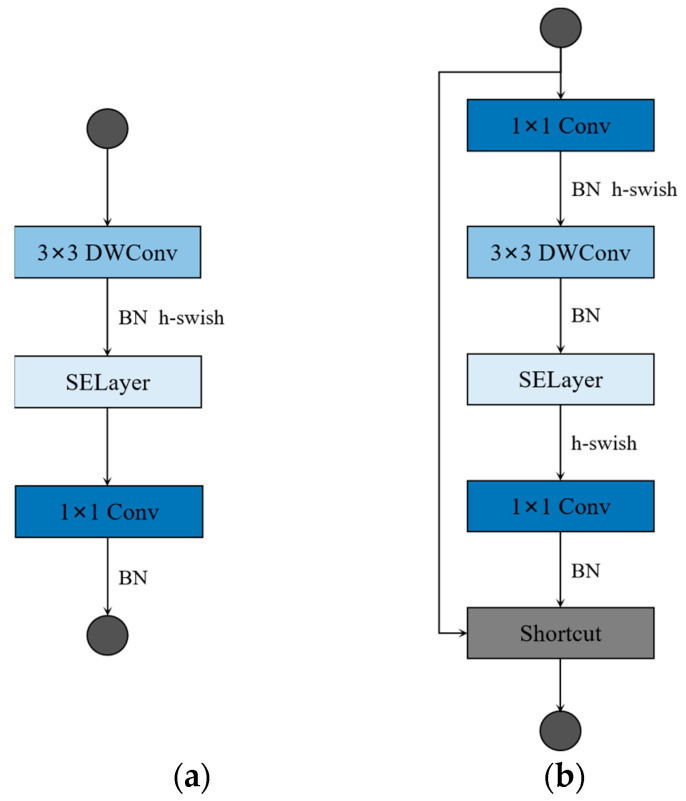
Building blocks of MobileNetv3. (**a**) No channel expansion units; (**b**) the unit of channel expansion with residual connection (when stride is set to one, and in_channels are equal to out_channels). DWconv denotes depthwise convolution.

**Figure 2 sensors-23-03268-f002:**
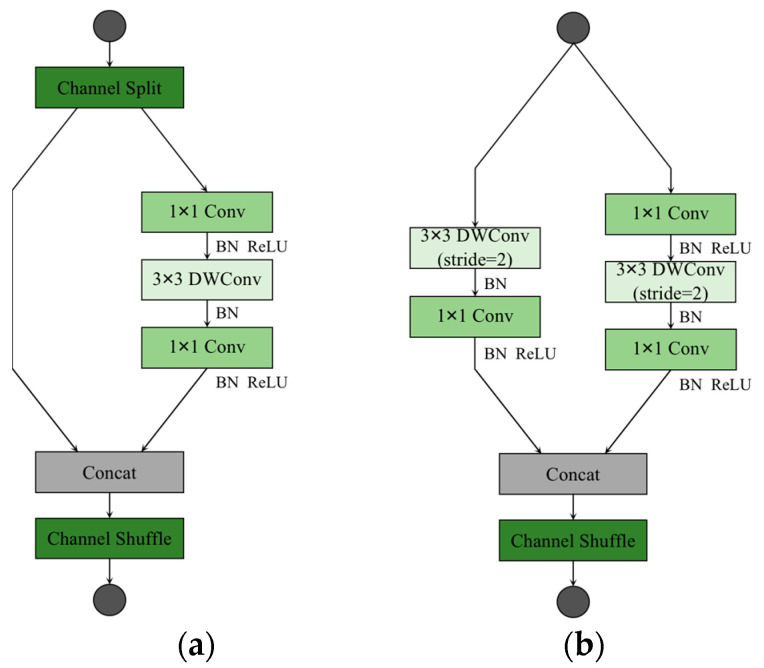
Building blocks of ShuffleNetv2. (**a**) The basic unit; (**b**) the spatial downsampling unit. DWConv denotes depthwise convolution.

**Figure 3 sensors-23-03268-f003:**
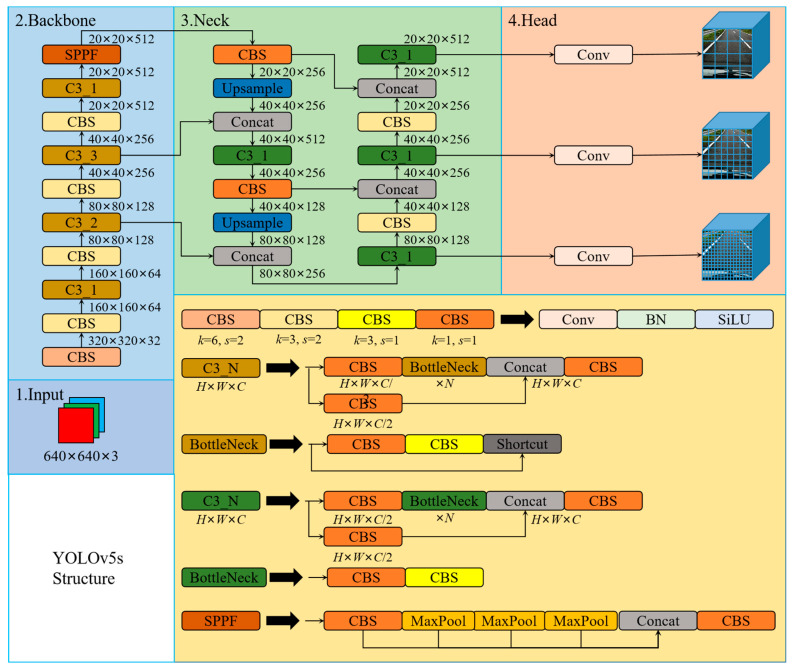
The structure of the YOLOv5-s.

**Figure 4 sensors-23-03268-f004:**
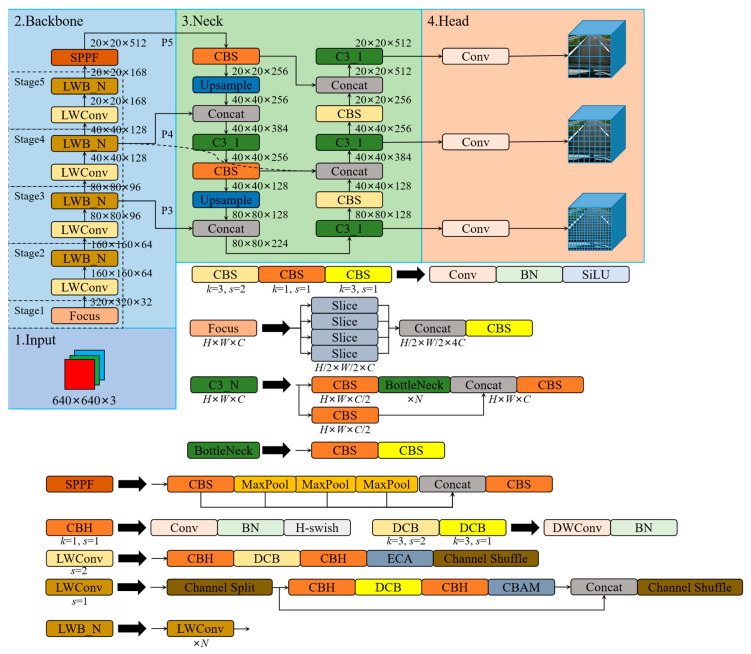
The network structure of the YOLO-LWNet-Small. Where, LWConv (*s* set to one) is the basic unit in the LWC module, and LWConv (*s* set to two) is the unit for spatial downsampling in the LWC module. In LWConv (*s* set to one), the CBAM attention module is used as attention. The ECA attention module is used as attention in LWConv (*s* set to two). In the small version, the hyperparameter *N* of LWblock (LWB) in stage two, three, four, and five in the backbone is set as one, four, four, and three, respectively.

**Figure 5 sensors-23-03268-f005:**
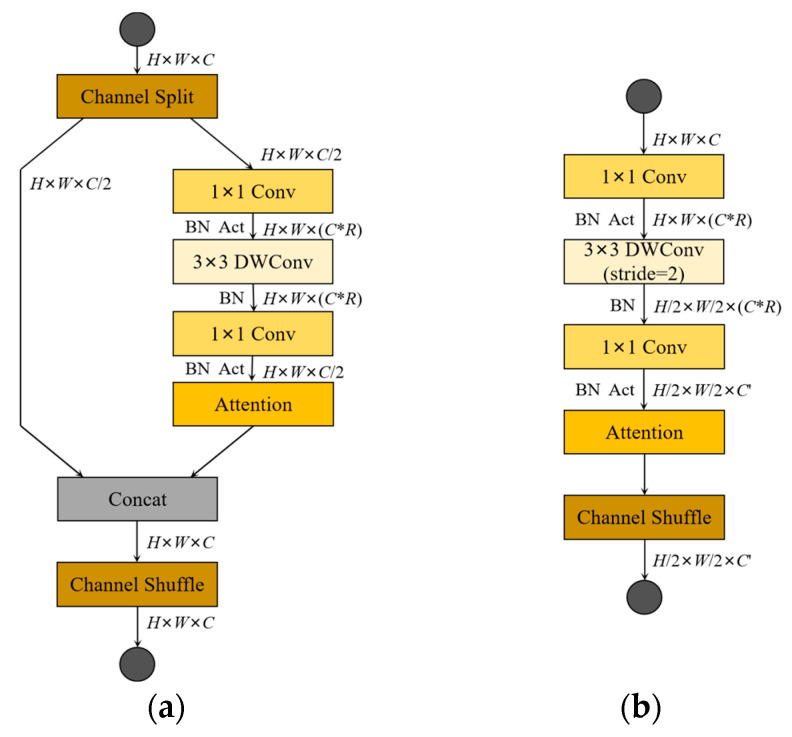
Building blocks of our work (LWC). (**a**) The basic unit (stride set to one); (**b**) the spatial downsampling unit (stride set to two). DWConv denotes depthwise convolution. *R* denotes expansion rate. Act denotes activates the function. Attention denotes the attention mechanism.

**Figure 6 sensors-23-03268-f006:**
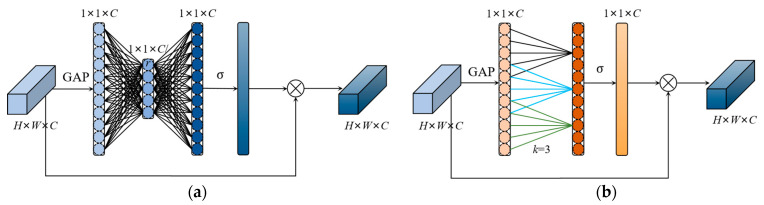
Diagram of SENet and ECANet. (**a**) Diagram of SENet; (**b**) diagram of ECANet. *k* denotes the convolution kernel size of 1D convolution in ECANet, and *k* is determined adaptively by mapping the channel dimension *C*. *σ* denotes the sigmoid function.

**Figure 7 sensors-23-03268-f007:**
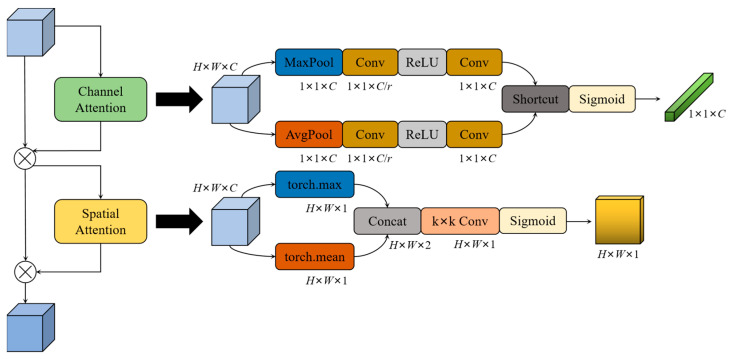
Diagram of CBAM. *r* denotes multiple dimension reduction in the channel attention module. *k* denotes the convolution kernel size of the convolution layer in the spatial attention module.

**Figure 8 sensors-23-03268-f008:**
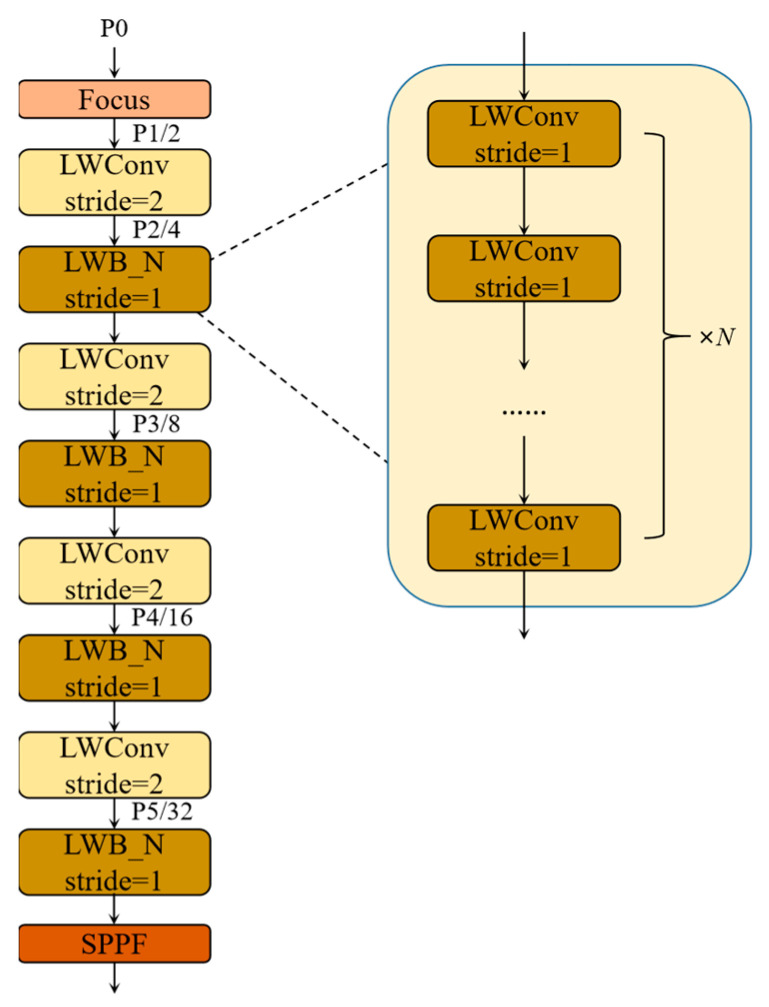
The overview of LWNet.

**Figure 9 sensors-23-03268-f009:**
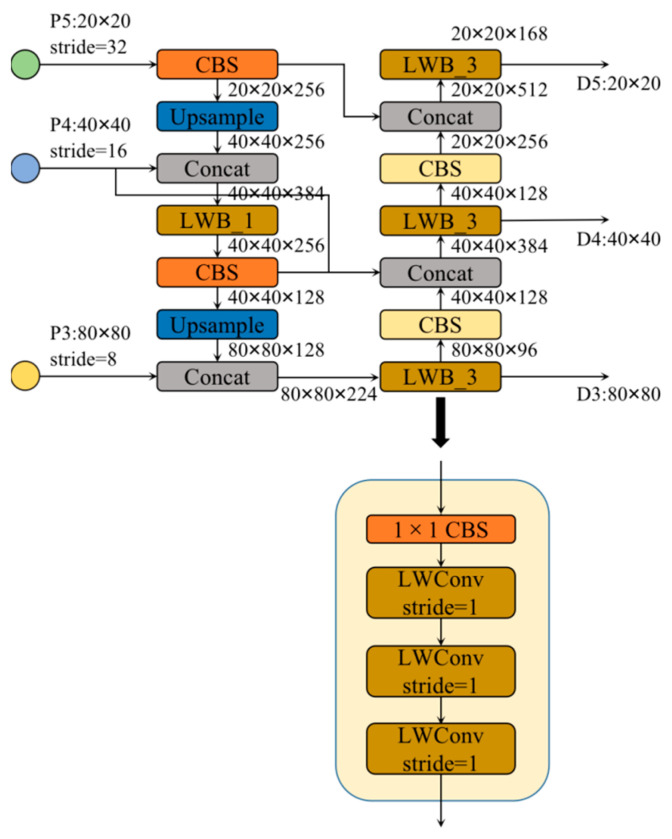
The overview of efficient feature fusion network.

**Figure 10 sensors-23-03268-f010:**
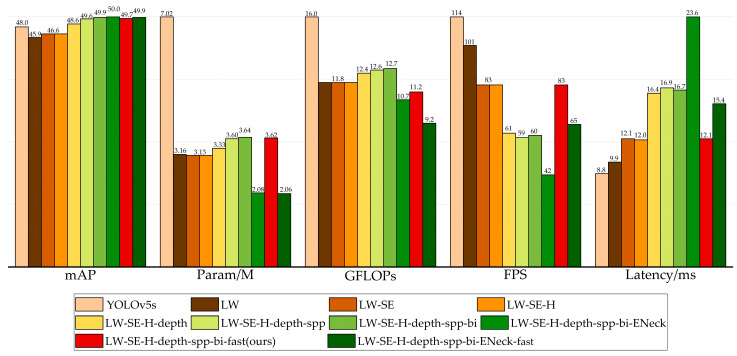
Ablation result of different methods on the test set.

**Figure 11 sensors-23-03268-f011:**
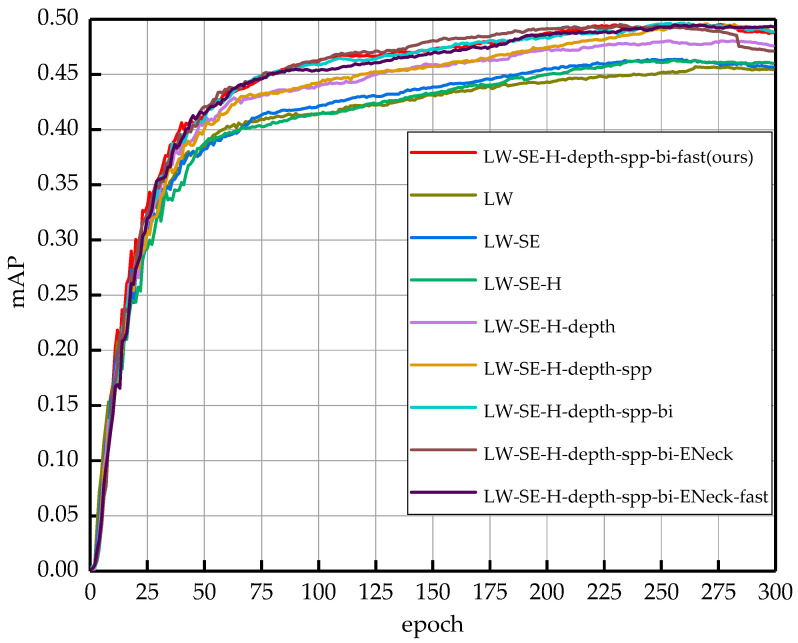
Performance comparison of the mean average precision using the training set of the RDD-2020. These curves represent the different improvement methods, as well as the final model structure.

**Figure 12 sensors-23-03268-f012:**
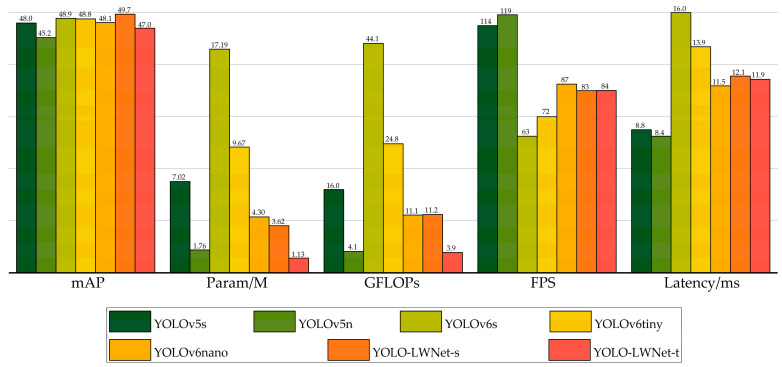
Comparison of different algorithms for object detection on the test set.

**Figure 13 sensors-23-03268-f013:**
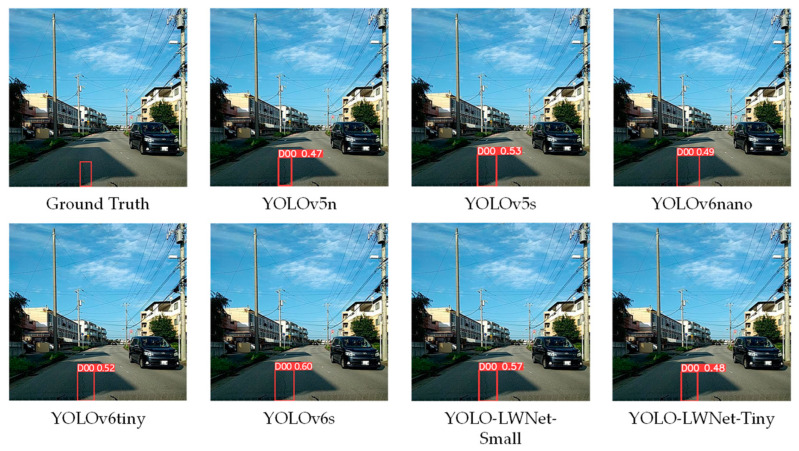
Comparison of the detection of longitudinal cracks (D00).

**Figure 14 sensors-23-03268-f014:**
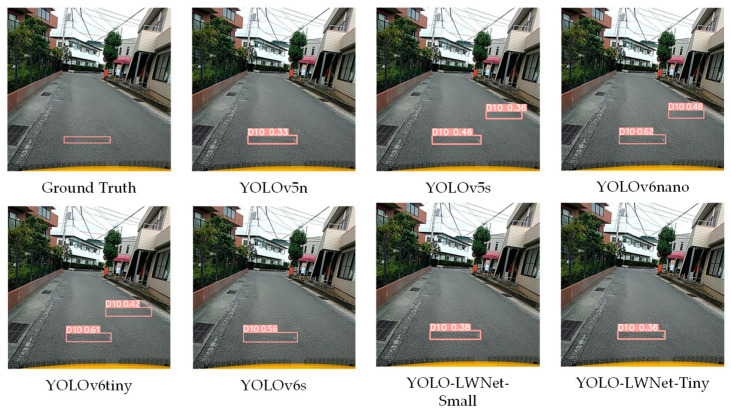
Comparison of the detection of transverse cracks (D10).

**Figure 15 sensors-23-03268-f015:**
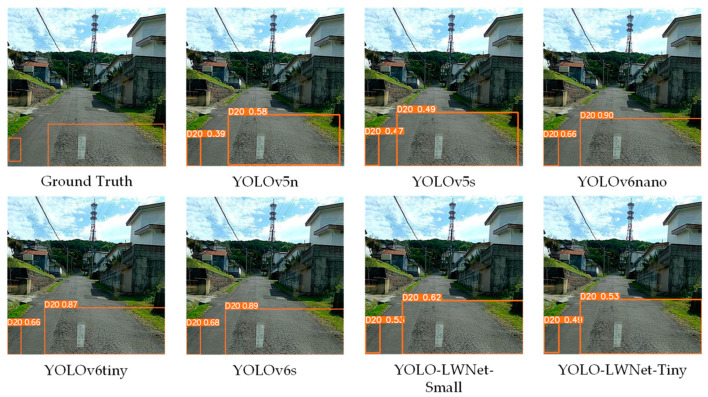
Comparison of the detection of alligator cracks (D20).

**Figure 16 sensors-23-03268-f016:**
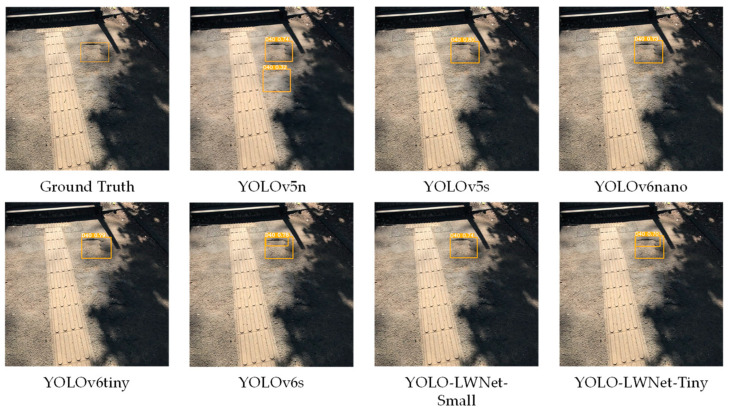
Comparison of the detection of potholes (D40).

**Figure 17 sensors-23-03268-f017:**
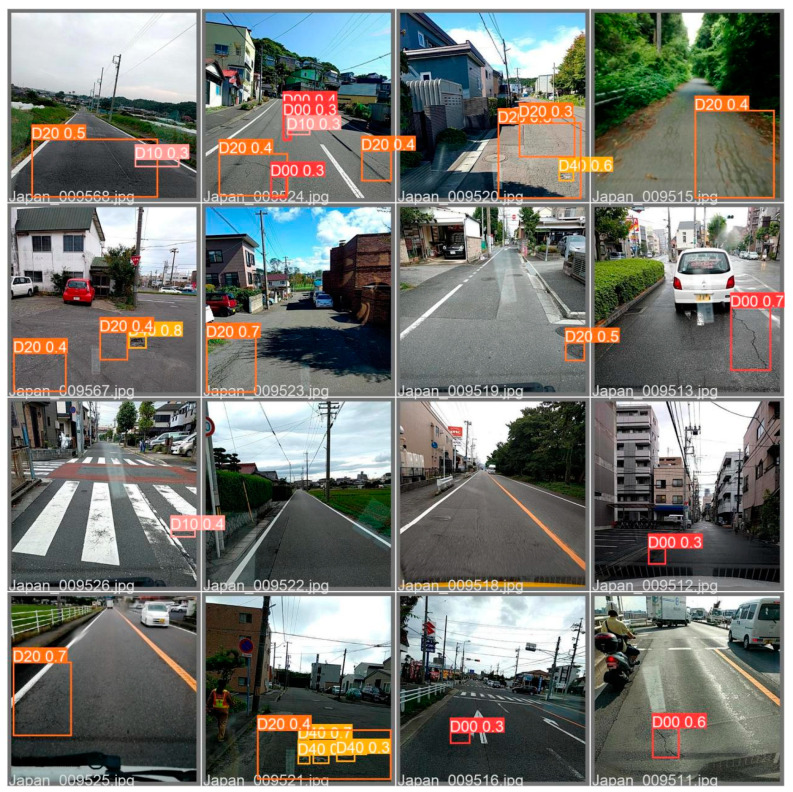
Pictures of detection results.

**Table 1 sensors-23-03268-t001:** Specification for a backbone—LWNet.

Operator	Output Size	s	act	n	Output Channels		Exp Channels	
					T	S	T	S
Image	640 × 640	-	-	-	3	3	3	3
Focus	320 × 320	-	-	1	16	32	-	-
LWConv	160 × 160	2	ECA	1	32	64	60	120
LWConv	160 × 160	1	ECA	1	32	64	60	120
LWConv	80 × 80	2	ECA	1	64	96	120	180
LWConv	80 × 80	1	ECA	4	64	96	120	180
LWConv	40 × 40	2	ECA	1	96	128	180	240
LWConv	40 × 40	1	ECA	4	96	128	180	240
LWConv	20 × 20	2	ECA	1	128	168	240	300
LWConv	20 × 20	1	ECA	3	128	168	240	300
SPPF	20 × 20	-	-	1	256	512	-	-

Each line describes one or more identical layers, repeated *n* times. Each of the same layers has the same number of output channels and exp channels (channels of the expansion layer). act denotes the attention mechanism. ECA denotes whether there is an efficient channel of attention in that block. *s* denotes stride. LWConv uses h-swish as the activation function. The main difference between the LWNet-Small and the LWNet-Tiny is the difference in output channels and exp channels.

**Table 2 sensors-23-03268-t002:** RDD-2020 dataset.

Class Name	Sample Image	Japan	India	Czech
D00	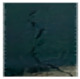	4049	1555	988
D10	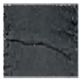	3979	68	399
D20	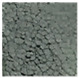	6199	2021	161
D40	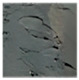	2243	3187	197

**Table 3 sensors-23-03268-t003:** Specification for the BLWNet-Small.

Operator	Output Size	s	SE	Output Channels	Exp Channels
Image	640 × 640	-	-	3	-
Focus	320 × 320	-	-	64	-
LWConv	160 × 160	2	√	64	120
LWConv	160 × 160	1	√	64	120
LWConv	80 × 80	2	√	96	180
LWConv	80 × 80	1	-	96	180
LWConv	80 × 80	1	√	96	180
LWConv	80 × 80	1	-	96	180
LWConv	40 × 40	2	√	128	240
LWConv	40 × 40	1	√	128	240
LWConv	40 × 40	1	√	128	240
LWConv	20 × 20	2	√	168	300
LWConv	20 × 20	1	√	168	300

SE denotes whether there is a Squeeze-And-Extract in that block, “√” indicates the presence of SE, otherwise it does not exist. *s* denotes stride. The swish nonlinear function is selected as the activation function in the LWConv module unit.

**Table 4 sensors-23-03268-t004:** Specification for the BLWNet-Large.

Operator	Output Size	s	SE	Output Channels	Exp Channels
Image	640 × 640	-	-	3	-
Focus	320 × 320	-	-	64	-
LWConv	160 × 160	2	-	64	120
LWConv	160 × 160	1	-	64	120
LWConv	80 × 80	2	√	96	180
LWConv	80 × 80	1	-	96	180
LWConv	80 × 80	1	√	96	180
LWConv	80 × 80	1	-	96	180
LWConv	80 × 80	1	√	96	180
LWConv	40 × 40	2	√	128	240
LWConv	40 × 40	1	√	128	240
LWConv	40 × 40	1	-	128	240
LWConv	40 × 40	1	√	128	240
LWConv	40 × 40	1	-	128	240
LWConv	40 × 40	1	√	128	240
LWConv	20 × 20	2	√	168	300
LWConv	20 × 20	1	-	168	300
LWConv	20 × 20	1	√	168	300

“√” indicates the presence of SE, otherwise it does not exist.

**Table 5 sensors-23-03268-t005:** Comparison of different lightweight networks as backbone networks.

Method	Backbone	*mAP*	Param/M	FLOPs	FPS	Latency/ms
YOLOv5	MobileNetV3-Small	43.0	3.55	6.3	93	10.7
	MobileNetV3-Large	47.1	13.47	24.7	80	12.5
	ShuffleNetV2-x1	42.8	3.61	7.5	89	11.2
	ShuffleNetV2-x2	45.1	14.67	29.7	83	12.1
	BLWNet-Small(ours)	45.9	3.16	11.8	101	9.9
	BLWNet-Large(ours)	48.2	11.30	27.3	83	12.1

**Table 6 sensors-23-03268-t006:** Different improvement schemes.

Scheme	BLWNet	CBAM/ECA	Hardswish	Depth	SPPF	BiFPN	ENeck	ECA
LW	√							
LW-SE	√	√						
LW-SE-H	√	√	√					
LW-SE-H-depth	√	√	√	√				
LW-SE-H-depth-spp	√	√	√	√	√			
LW-SE-H-depth-spp-bi	√	√	√	√	√	√		
LW-SE-H-depth-spp-bi-ENeck	√	√	√	√	√	√	√	
LW-SE-H-depth-spp-bi-fast	√		√	√	√	√		√
LW-SE-H-depth-spp-bi-ENeck-fast	√		√	√	√	√	√	√

“√” in the table indicates the presence of the corresponding module.

## Data Availability

Data are contained within the article.

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
