# Peer review of "YOLO-LWNet: A Lightweight Road Damage Object Detection Network for Mobile Terminal Devices"

_sensors, 2023, doi:10.3390/s23063268_

Round 1

Reviewer 1 Report

1.This paper is an improvement on the subject of yolov5, whose main network structure is used for YOLOv5, replacing parts of it with mobilenet structures, so the algorithm name using RDD-mobilenet may not be appropriate.

2.The presentation of the dataset is inadequate and a picture of each defect could be presented and defined to help the reader identify the accuracy of the results.

3.The training process is not reflected, including its loss curve and AP curve, and the confidence level in the final result  is low, has the model reached convergence?

4.The comparison of different algorithms does not give the comparative detection results separately, making it difficult to effectively illustrate the effectiveness of the algorithms.

Reviewer 2 Report

1. The abstract should be concise and comprehensive, not too lengthy.

2. The abbreviations like GERPHO, LCPC, and all others should be defined first and then used.

3. What is the origin of the formula (1, 5, 6, 7)? Reference should be added.

4. How the formulae are used to calculate the results presented in table 1?

5. From where the data is obtained that is presented in table 2? Add the reference. 

6. How the results are validated? A comparison to the existing literature should be added.

Reviewer 3 Report

The comments are mentioned in the attached file.

Reviewer 4 Report

This manuscript (ID: sensors-2249319) investigates a light road damage detection network, where different restrictions are stated.  Small and micro RDD mobile networks are introduced. According to the content presented, the following aspects need to be improved.

1. How to distinguish between damage and debris accumulation on the road?  For example, on the way to the right of the first row in Figure 10, there are weeds and branches on the road, or there are other obstacles on the road surface in the urban road.

2. Can the mathematical analysis of lightweight network building blocks provide more expressions?

3. The conclusion needs to be succinctly extracted into about three small paragraphs, such as the important results of this paper, the improvement of detection accuracy or important application value. 

Round 2

Reviewer 1 Report

It was modified as required, but there is still a problem in the modified version : the clarity of the label in the pictures compared by different algorithms is different and needs to be unified.

Reviewer 4 Report

In this manuscript (ID: sensors-2249319), the author made a comprehensive revision of the paper. Relevant discussions in the references are more substantial. The discussion of data results is also more detailed.

1. In line 181, the abbreviations of DW and PW are not consistent with the initial letters of the previous words. Please explain why. 
